# Preemptive Venoarterial Extracorporeal Membrane Oxygenation for Liver Transplantation—Judicious Candidate Selection

**DOI:** 10.3390/jcm12154965

**Published:** 2023-07-28

**Authors:** Jennifer Lee, Wesley L. Allen, Courtney L. Scott, Stephen Aniskevich, Sher-Lu Pai

**Affiliations:** Department of Anesthesiology and Perioperative Medicine, Mayo Clinic, Jacksonville, FL 32224, USA; lee.jennifer3@mayo.edu (J.L.); scott.courtney@mayo.edu (C.L.S.); aniskevich.stephen2@mayo.edu (S.A.); pai.sherlu@mayo.edu (S.-L.P.)

**Keywords:** pulmonary hypertension, liver transplantation, extracorporeal membrane oxygenation, patient selection, surgical candidacy, cardiovascular disease, clinical decision making, critical care/intensive care management, liver (allograft) function/dysfunction

## Abstract

Portopulmonary hypertension is a relatively common pathologic condition in patients with end-stage liver disease. Traditionally, severe pulmonary hypertension is regarded as a contraindication to liver transplantation (LT) due to a high perioperative mortality rate. Recently, extracorporeal membrane oxygenation (ECMO) has been utilized for intraoperative management of LT. As venoarterial (VA) ECMO may benefit certain high-risk LT patients by reducing the ventricular workload by the equivalent of the programmed flow rate, its usage requires multidisciplinary planning with considerations of the associated complications. We highlighted two cases at our single-center institution as examples of high-risk pulmonary hypertension patients undergoing LT on planned VA ECMO. These patients both survived the intraoperative period; however, they had drastically different postoperative outcomes, generating discussions on the importance of judicious patient selection. Since ECMO has removed the barrier of intraoperative survivability, the patient selection process may need to put weight on the patient’s potential for postoperative recovery and rehabilitation. Considerations on LT recipients undergoing preemptive ECMO need to expand from the ability of the patients to withstand the demands of the surgery during the immediate perioperative period to the long-term postoperative recovery course.

## 1. Introduction

Portopulmonary hypertension (PPH) is a relatively common pathologic condition in patients with end-stage liver disease (ESLD). PPH is defined as a mean pulmonary artery pressure (mPAP) greater than 25 mmHg and pulmonary capillary wedge pressure (PCWP) less than 15 mmHg in the setting of portal hypertension [1]. The additional criteria of pulmonary vascular resistance (PVR) over 120 dynes·s cm^−5^·is occasionally included [2]. It has been reported that, for liver transplant (LT) patients, intraoperative and postoperative mortality is directly linked to the severity of pulmonary hypertension [3]. LT patients with a mPAP of 35–50 mmHg carry a 50% perioperative mortality and those with a mPAP of greater than 50 mmHg carry a 100% perioperative mortality [3]. Traditionally, severe pulmonary hypertension with a mPAP of 50 mmHg or higher is regarded as a contraindication to LT due to a high perioperative mortality rate.

The time of donor graft reperfusion is often considered the most challenging stage of the LT surgery since it is associated with significant hemodynamic instability and electrolyte abnormalities due to the release of inflammatory mediators, vasoactive substances, potassium, and hydrogen ions from the ischemic donor graft into the systemic circulation. During the reperfusion stage, the heart is under enormous stress with added risks for cardiac arrhythmias, profound hypotension, direct myocardial depression, and acute pulmonary artery hypertension [4]. Extracorporeal membrane oxygenation (ECMO) is a mechanical circulatory pump with oxygenation and ventilation capabilities that can be utilized for hemodynamic support during this time. Venoarterial (VA) ECMO is used in patients who require cardiac and respiratory support with the circuit connected in parallel to the heart and lungs [5], whereas venovenous (VV) ECMO is limited to patients with respiratory failure only with the circuit connected in series to the heart and lungs [5]. In VA ECMO, blood is removed from the venous cannula, pressurized via the pump, and directed through the oxygenator prior to outflow into the arterial cannula [5]. While outside the body, hemoglobin is fully saturated with oxygen, and carbon dioxide is removed by the membrane oxygenator. Oxygenation is determined by the flow rate while carbon dioxide elimination is controlled by adjustment of countercurrent gas flow through the oxygenator [5].

As VA ECMO reduces the ventricular workload by the equivalent of the programmed flow rate, it can provide significant support for the high-risk LT population during the intraoperative period. Over the past several decades, ECMO has been used to treat acute pulmonary and cardiac dysfunction both during and after LT [4]. Preemptive initiation of VA ECMO, as opposed to rescue therapy, has allowed LT patients who were previously deemed nonsurgical candidates to survive the intraoperative period by significantly decreasing the risk of acute right ventricular failure, especially during the reperfusion stage [4]. In this single-center case series, we highlighted two cases as examples of high-risk pulmonary hypertension patients undergoing LT on planned VA ECMO at our institution. A standardized liver transplantation protocol created by our institution is commonly used and adapted to each individual case. These two cases followed our typical liver transplant protocol with several modifications to accommodate the use of ECMO intraoperatively. Both cases were performed with invasive monitoring including an arterial line, central venous catheter, pulmonary artery catheter, and TEE. The ECMO flows were adjusted under TEE guidance and based on the patient’s hemodynamics intraoperatively. Both of these patients survived the perioperative period; however, they had drastically different postoperative outcomes, generating discussions on the importance of judicious patient selection.

## 2. Case Description

### 2.1. Case 1

A 49-year-old female with ESLD secondary to cryptogenic cirrhosis and PPH presented to our institution in January 2022 for LT. The patient’s PPH was diagnosed via serial TTEs during her hospital course and by right heart cath as part of a routine pretransplant workup. She was diagnosed with ESLD when in 2021 a new mass was discovered to be hepatocellular carcinoma. Soon after, she began developing ascites and jaundice and was referred to our institution for a liver transplant workup. She initially presented with a model for end-stage liver disease (MELD) score of 19 and a pretransplant transthoracic echocardiogram (TTE) revealed a mildly enlarged right ventricle (RV) with normal systolic function, an estimated right ventricular systolic pressure (RVSP) of 39 mmHg, a mild to moderate intrapulmonary shunt, a small patent foramen ovale, and a left ventricular ejection fraction of 73% (refer to Figure 1). The patient’s preoperative functional capacity was poor with the patient and family member reporting that she had difficulty performing activities of daily living and experienced significant dyspnea walking around the house.

When a suitable donor organ was available at the end of January, the patient was brought to the operating room. After the induction of general anesthesia and the placements of a transesophageal echocardiogram (TEE) and a pulmonary artery catheter (PAC), the pulmonary artery pressure (PAP) of 89/34 mmHg and mPAP of 57 were discovered. While the patient remained hemodynamically stable, her oxygen saturation ranged from 85 to 92 percent. The PAP was significantly higher than previously shown on preoperative TTE. The risk of intraoperative mortality was deemed too high, and the transplant was aborted. The patient was subsequently transported to the intensive care unit (ICU) where she was started on sildenafil, extubated, transferred to the floor, and discharged home.

Two weeks later, in mid-February, the patient presented to the emergency department for worsening dyspnea. A repeat TTE showed worsening pulmonary hypertension with an estimated RVSP of 85 mmHg, a mPAP of 56 mmHg, moderate tricuspid regurgitation, and a moderately enlarged right ventricular chamber size with borderline contractile dysfunction. Due to the severity of symptoms and lack of curative options, multidisciplinary discussions concluded the plan for the patient to proceed with LT on preemptive VA ECMO support at the end of February. Twenty-four hours prior to the LT, the patient was placed on VA ECMO with a 23 French right internal jugular venous cannula, a distal cannula tip advanced to the infra-hepatic inferior vena cava (IVC), and a 21 French right axillary arterial cannula. ECMO support was initiated, achieving 2.5 L/min of flow. The patient’s PAP significantly improved from systolic in 70 mmHg to approximately 50 mmHg. The day following ECMO cannulation, a liver became available, and the patient returned to the operating room for transplantation. The intraoperative anesthetic plan included general anesthesia and peripheral VA ECMO. Radial and brachial arterial lines, a central venous line with PAC, and a hemodialysis catheter were placed. Physicians from Transplant Anesthesiology, Cardiac Anesthesiology, Transplant Surgery, and Cardiothoracic Surgery teams were present throughout the procedure. TEE was used to monitor cardiac function and guide ECMO flows throughout the procedure. VA ECMO flows were maintained at a minimum of 1.5 L/min to minimize clotting and serial activated clotting times (ACT) were monitored every 30 min and maintained between 170 to 220 s. No anticoagulation was required throughout the case due to the patient’s baseline coagulopathy. The patient was kept on a low-dose norepinephrine infusion (less than 0.05 mcg/kg/min) and 0.04 units/min of vasopressin infusion throughout the procedure. Intraoperative TEE on VA ECMO revealed a severely enlarged right ventricle, borderline reduced right ventricular systolic function, patent foramen ovale, mild tricuspid regurgitation, and normal left ventricular ejection fraction. Immediately following reperfusion, PAP significantly increased from a systolic of 40 mmHg to 78 mmHg, with TEE showing severe right ventricular systolic dysfunction and impending RV collapse (refer to Figure 2). VA ECMO flows were increased from 1.6 to 2.4 L/min to assist the RV, in lieu of inotropic support, with improvement in right ventricular systolic function and eventual normalization. Intravenous calcium chloride (2 g) was given for reperfusion with no further intraoperative need for inotropic or vasopressor support. The patient remained hemodynamically stable throughout the remainder of the procedure. Prior to leaving the operating room, TEE images noted normal right ventricular systolic function, unchanged moderate to severe right ventricular dilation, and preserved left ventricular ejection fraction. The patient was then transported to the ICU on VA ECMO with inhaled nitric oxide, inotropic, and vasopressor support, including norepinephrine at 0.04 mcg/kg/min and vasopressin at 0.04 units/min. On postoperative day 4, the patient was deemed ready for VA ECMO decannulation and returned to the OR for decannulation. Unfortunately, her postoperative course was mired with complications including worsening PAP, continued hemodynamic instability necessitating inotropic and vasopressor support, worsening hypoxia, and severe coagulopathy necessitating several transfusions. The patient ultimately expired in the ICU 6 weeks post LT at the beginning of March 2022 from refractory hypoxia and hypotension leading to asystole.

### 2.2. Case 2

A 58-year-old male with ESLD secondary to cryptogenic cirrhosis, status post-Ross (pulmonary autograft) procedure, and prosthetic pulmonic valve placement presented to our institution in February 2022 with an estimated right ventricular systolic pressure (RVSP) of 61 mmHg and associated chamber dilatation. He was diagnosed with cirrhosis around 2015–2016 when he was found to have low platelets during routine labs. The patient’s PPH was diagnosed via serial TTEs during his hospital course and by right heart cath as part of a routine pretransplant workup. The patient was initially evaluated for LT several years prior, but his case was postponed due to pulmonary valve stenosis necessitating transcatheter pulmonary valve implantation. At the end of February, he was admitted for decompensated cirrhosis with his pre-transplant hospitalization course complicated by acute hypoxic respiratory failure and gastrointestinal bleeding, requiring transjugular intrahepatic portosystemic shunt revision. Pre-transplant TTE showed a pulmonary valve prosthetic gradient of 19 mmHg, an estimated RVSP of 61 mmHg, a mild–moderately enlarged right ventricular chamber size with preserved systolic function, and a normal left ventricular ejection fraction. A multidisciplinary committee met prior to LT to discuss the specific intraoperative risks associated with the patient’s comorbidities. Due to the high risk for right ventricular collapse, the plan of preemptive VA ECMO for intraoperative support was formulated.

When a liver became available in late March of 2022, the intraoperative anesthetic management and invasive monitors included general anesthesia, peripheral VA ECMO, bilateral radial arterial lines, venous access central line with a PAC, and a rapid infusion catheter to allow intraoperative continuous renal replacement therapy (CRRT). Physicians from Transplant Anesthesiology, Cardiac Anesthesiology, Transplant Surgery, and Cardiothoracic Surgery teams were present throughout the procedure. After the incision, 2000 units of intravenous heparin was administered in anticipation of ECMO cannulation to prevent thrombotic complications. A 19 French arterial cannula was placed into the right common femoral artery, a 23 French venous cannula was placed into the left common femoral vein, and a 6 French distal perfusion cannula was placed by cardiac surgery. ECMO support was initiated and maintained with flows ranging from 1.9 to 2.4 L/min. Serial ACT measurements were drawn every 30 min to an hour with the therapeutic goal of 170 to 200 s. Appropriate ACT measurements were maintained throughout the procedure. Intraoperative TEE on VA ECMO showed a moderate–severely enlarged right ventricle with normal systolic function, a moderate tricuspid valve regurgitation, a moderate–severe pulmonary valve stenosis with a peak gradient of 37 mmHg, and a preserved left ventricular ejection function. At reperfusion, acute right-sided chamber dilatation and dysfunction with concomitant atrial septal bowing, right ventricular distention, and biphasic ventricular septal flattening on TEE showed right ventricular volume and pressure overload with reduced left ventricular filling. Intravenous calcium chloride (500 mg) as inotropic support was given with ECMO flows maintained at 2 L/min. Right ventricular function improved, and the patient remained hemodynamically stable through reperfusion with only a low dose norepinephrine infusion at 0.03 mcg/kg/min running for support. Right ventricular function returned to baseline 5 min post reperfusion. The patient remained hemodynamically stable throughout the procedure. The ECMO support was subsequently discontinued, and the patient was decannulated prior to abdominal closure. Post VA ECMO decannulation TEE exam showed moderate right ventricular chamber dilation with normal systolic function. No complications were noted with caval anastomosis or increased hemorrhage. The patient was transferred to the ICU postoperatively hemodynamically stable without the need for ECMO, inotropic, or vasopressor support. The day after the LT, mechanical ventilation and CRRT were both discontinued while the patient continued to have no need for vasopressors and inotropes for hemodynamic support. The major postoperative complications due to the ECMO cannulas were narrowing at the venous piggyback anastomosis. This was corrected by balloon angioplasty. The rest of his postoperative course was unremarkable with no other significant complications and good graft function. He was discharged to outpatient rehab at the end of April 2022 after 24 days of hospitalization. This patient was seen one year post transplant in the outpatient setting by the medical transplant team. Both patient and graft are doing well as evidenced by labs and ultrasound.

## 3. Results

These two cases at our single-center institution present the successful intraoperative management of LT with a high risk of right ventricular failure due to severe pulmonary hypertension or equivalent. Although both patients survived the immediate perioperative period with minimal resuscitation by utilizing VA ECMO support through the LT surgery, principally during the reperfusion stage, their long-term postoperative outcomes greatly differed, highlighting the importance of appropriate patient selection for this strategy. The first patient was a poor surgical candidate due to her poor preoperative functional capacity, frail condition, severe pulmonary hypertension, and right ventricular systolic dysfunction. With her inability to complete basic activities of daily living, her functional capacity was estimated to be one metabolic equivalent. While her PPH was presumed reversible, the patient was debilitation and unable to make a meaningful recovery postoperatively. The patient in the second case presented a more favorable postoperative rehabilitation potential due to his adequate preoperative functional capacity of approximately four metabolic equivalents with a normal right ventricular systolic function. In the second case, ECMO was used to address the immediate perioperative cardiopulmonary risk.

## 4. Discussion

PPH is a common, but serious, sequelae of ESLD with high mortality implications for LT. Without right ventricular unloading and support via intraoperative ECMO, this patient population would be unlikely to survive LT [6]. VA ECMO has provided intraoperative hemodynamic support to LT patients that were previously deemed not transplantable due to hemodynamic instability. It is particularly effective in the case of cardiopulmonary failure with a reversible cause and not due to an underlying chronic condition. In its most basic form, VA ECMO reduces the workload of the heart by the magnitude of the programmed cardiac output. ECMO flows are adjusted under TEE evaluation to decompress the RV and avoid acute collapse during reperfusion. Mechanical support should reduce the need for excessive inotropic and vasopressor support, particularly during this stage. Special attention to maintaining systolic blood pressure below 140 mmHg should be considered to mitigate the risk of bleeding at the arterial cannulation site. Attention must also be paid to the venous pressure in the admission cannula to avoid venous pressures above 100 mmHg due to the risk of air entrainment. Venous pressures over 60 mmHg signify hypovolemia and must be promptly addressed with volume resuscitation. The safety and logistics of such an approach require multidisciplinary coordination between Anesthesiology, Cardiothoracic Surgery, Cardiology, Hepatology, Transplant Surgery, Critical Care, and Perfusion teams. Although VA ECMO has the benefit of hemodynamic support, strategic planning on risk mitigations prior to surgery (Table 1) and establishing an ECMO management algorithm (Table 2) may be beneficial. Our institution has created a liver transplant protocol which we used with several modifications to accommodate intraoperative ECMO. When making the decision on the implementation of ECMO, factors including, but not limited to, cannulation site, anticoagulation, air entrainment into the circuit, thrombotic complications, and significant volume shifts, should be considered (Table 3). Regarding VA ECMO cannulation, the patient’s volume status should be optimized with euvolemia as the goal. When the arterial inflow cannula provides retrograde flow via the descending aorta, afterload reduction is important to ensure distal perfusion of the lower body [4]. Postoperative ICU care must also be discussed prior to surgery. The timing of ECMO decannulation requires a multi-disciplinary approach to assess the risks of continued anticoagulation/hemostasis versus cardiopulmonary stability at variable ECMO flows under comprehensive TEE guidance. The patient is progressively weaned off ECMO in the ICU and a trial-off period is performed to prove that the patient is able to maintain life support through conventional means. Since these two patients were NOT dependent on ECMO prior to transplantation but were instead cannulated to support their cardiac function during the extreme stress of liver transplantation, ECMO decannulation was able to be conducted within a few days after surgery once the stressor had been removed. A repeat echocardiography is also performed to ensure adequate cardiac recovery and function prior to decannulation.

While some complications are associated with the usage of ECMO [5], LT patients present with additional challenges for ECMO usage. Immunosuppression, a necessity to prevent organ rejection, has been shown to increase the rate of ECMO-related mortality [3]. Coagulopathy is another common condition found in patients with ESLD. Anticoagulation usage in ECMO can increase bleeding in an already sanguinary procedure and coagulopathic LT patient population. However, no anticoagulation can result in circuit thrombosis and intra-cardiac thrombosis. Intraoperative anticoagulation should be discussed on a case-by-case basis with constant intraoperative reevaluation via serial ACT measurements and a minimal ECMO flow of 1 L/min to prevent in-line thrombosis. In the literature, low-dose unfractionated heparin (40 mcg/kg) or even holding all anticoagulation has not been shown to precipitate any coagulation-specific adverse effects [7]. One of the most devastating thrombotic complications is intra-cardiac thrombosis, whose clinical manifestation can range anywhere from silence to complete cardiovascular collapse. As reported by Peiris et al., left-sided intracardiac thrombosis is associated with high mortality [12]. Treatment protocols for this complication should be established prior to surgery to allow for prompt diagnosis and treatment. Small intra-cardiac clots which do not generate hemodynamic instability may require no treatment or incremental dosages of heparin [13], while intracardiac thrombosis causing hemodynamic instability and cardiopulmonary collapse may require immediate treatment with alteplase [13].

The preemptive use of ECMO for high-risk pulmonary hypertension patients for liver transplantation has been supported in the literature. A Duke case report presents a successful liver transplantation on a high-risk pulmonary hypertension patient with preemptive initiation of VA-ECMO after previously aborted attempts [6]. A high-volume transplant center with an extensive database of liver transplant patients on ECMO present ECMO as a viable rescue strategy that should be considered in patients with severe cardiopulmonary compromise [5]. As ECMO now provides intraoperative hemodynamic support to withstand the demands of LT surgery, the focus may need to be redirected onto the long-term postoperative recovery course. Frailty is a strong, independent factor associated with increased mortality in patients with cirrhosis [14]. In addition to measuring functional capacity via metabolic equivalents, several frailty measurement tools are available with promising predictability on postoperative 90-day mortality, hospital length of stay, and rehabilitation potential for LT patients [15] (Figure 3, Table 4). These tools may be utilized during the patient selection process, estimating the long-term survival rates of patients with PPH if preemptive ECMO is planned for LT.

## 5. Conclusions

ECMO is a feasible option in high-risk patients. Multidisciplinary planning and careful consideration of associated complications with ECMO during LT is paramount. This case report presents the unique employment of VA ECMO as preemptive support during the intraoperative period. The two different postoperative outcomes demonstrate the importance of judicious candidate selection with a new focus on the patient’s long-term survivability and rehabilitation potential in addition to the immediate intraoperative survivability.

## Figures and Tables

**Figure 1 jcm-12-04965-f001:**
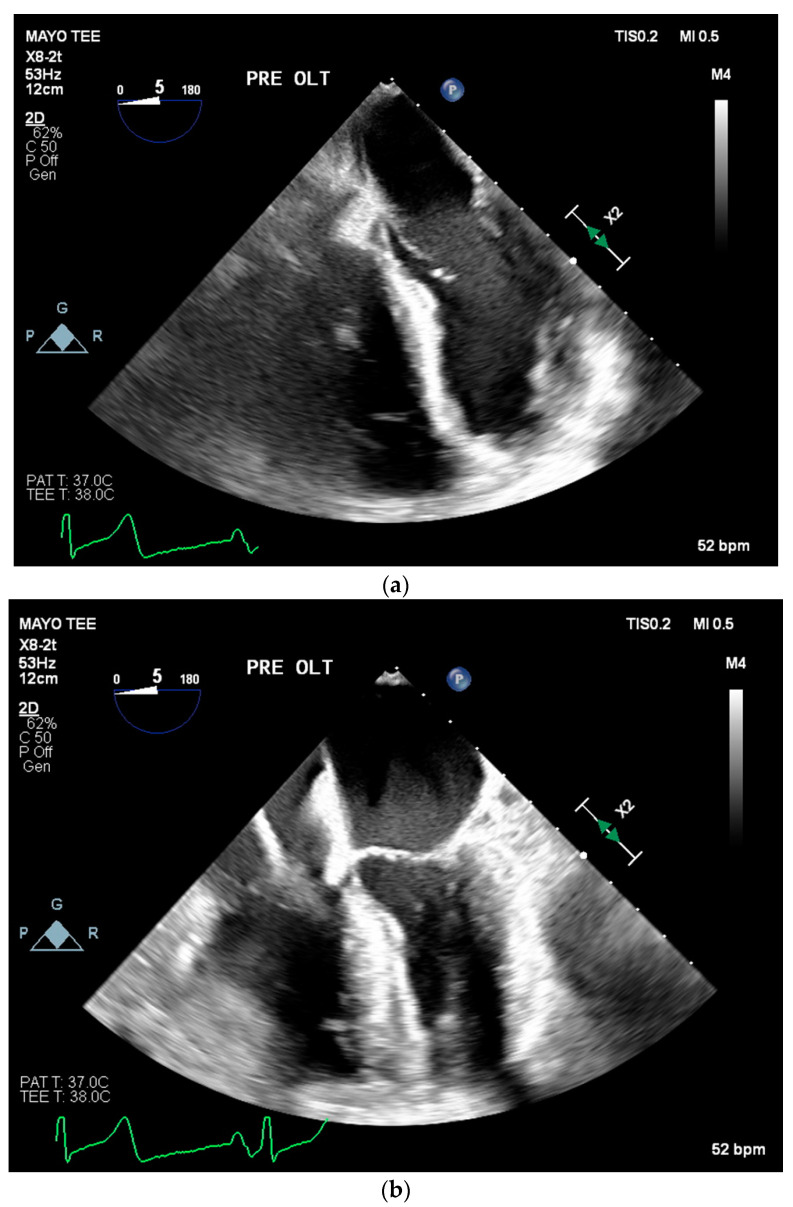
Intraoperative transesophageal echocardiogram of Patient 1. Baseline intraoperative transesophageal echocardiogram intraoperatively at the end of diastole (**a**) and systole (**b**) prior to liver transplantation showing a severely enlarged right ventricular chamber size, borderline reduced right ventricular systolic function, and left ventricular ejection fraction of 63%.

**Figure 2 jcm-12-04965-f002:**
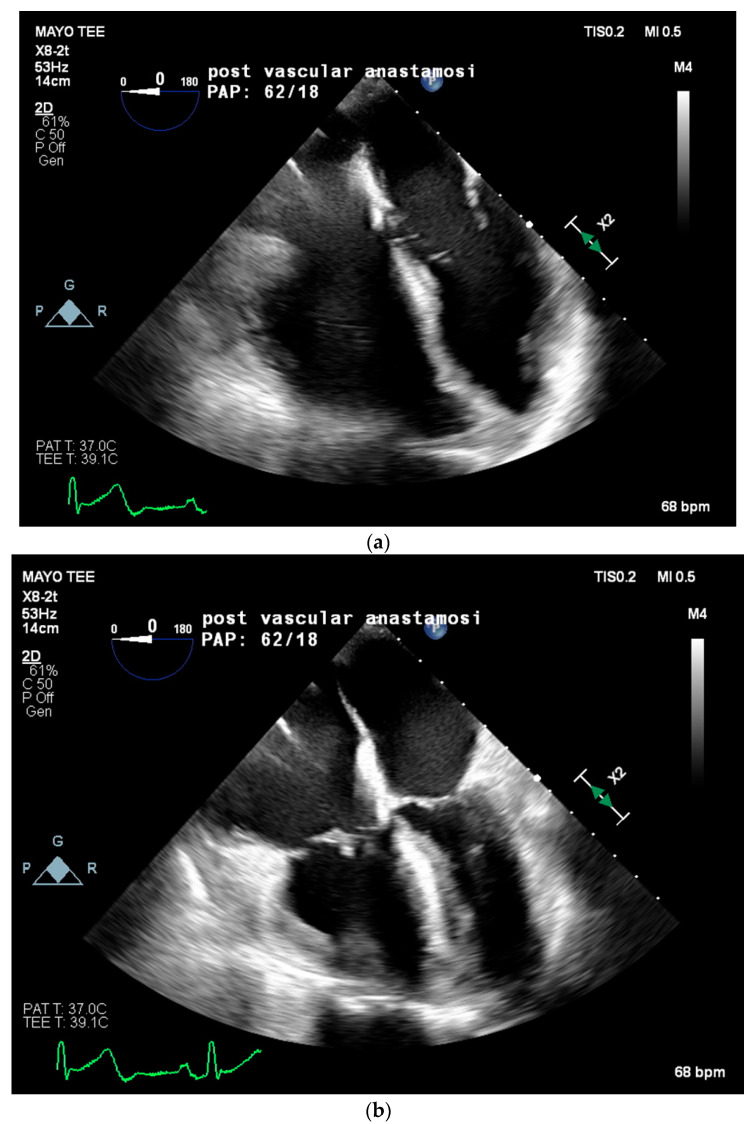
Following cross-clamp removal on reperfusion, pulmonary artery pressures significantly increased from a PA pressure of 40 mmHg to 78 mmHg with severe right ventricular systolic dysfunction. Images taken during diastole (**a**) and systole (**b**) are shown to appreciate the degree of right ventricular dysfunction present immediately following reperfusion. Septal bowing into the left ventricle from severe right-sided pressure overload can be appreciated in systole (**b**).

**Figure 3 jcm-12-04965-f003:**
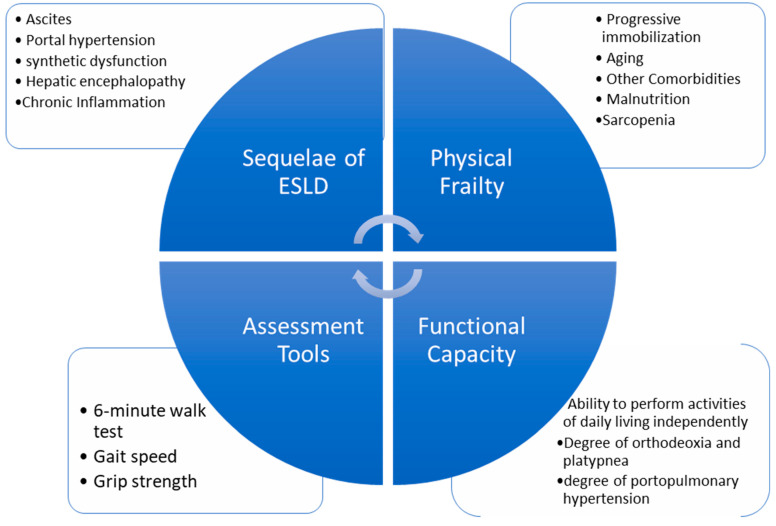
Components of frailty in end-stage liver disease [16,17].

**Table 1 jcm-12-04965-t001:** Advantages and disadvantages of ECMO [5,7].

Advantages	Disadvantages
Cardiopulmonary support decreasing workload of the heart	Cannula placement may interfere with venous clamping during the hepatic phase
Decreased inotropic and vasopressor support requirement	Thrombotic risk if flow too low or no anticoagulation used
Decreased ventilator support requirement *	Bleeding risk if anticoagulation used
Easily titratable flow rate and sweep	Significant volume shifts affect flow
	Significant multidisciplinary coordination required for implementation
	Risk of air entrainment into the circuit

* On peripheral VA-ECMO, especially at reduced flow, it is important to maintain ventilation to avoid “north-south syndrome” (Harlequin Syndrome). Dual circulation, or “north-south syndrome” is a phenomenon of unequal oxygenation between the upper and lower body when there is mixing of oxygenated blood returning into the femoral artery (from the ECMO circuit) and deoxygenated blood from the native pulmonary blood flow. In these cases of reduced flow, the ventilatory requirements remain unchanged.

**Table 2 jcm-12-04965-t002:** Protocol for ECMO during liver transplant.

Hemodynamic Goals	Reduce inotropic and vasopressor support, especially during reperfusion Adjust ECMO flows according to TEE Maintain SBP between 120–140 mmHg prior to reperfusion Keep SBP < 140 mmHg to decrease bleeding risk at arterial cannulation site P_venous_ magnitude is inversely equivalent to volume status. P_venous_ < 60mmHg magnitude represents hypovolemia Cardiac output measured by pulmonary artery catheter does not include additional VA ECMO output
Anticoagulation	Anticoagulation to be discussed on a case-by-case basis Intraoperative ACT measurements q 30 min Goal ACT: 170-200 s Minimum required ECMO flows to prevent thrombosis: 1 L/min
VA ECMO Cannulation Sites	Arterial cannulation: right subclavian vs. femoral artery Venous cannulation: right internal jugular (IJ) vs. femoral vein Ideal location of distal cannula tip is in the infra-hepatic IVC Cannula in Right IJ will traverse area where clamp is applied for piggyback procedure Placement in intrahepatic IVC decreases risk of air embolism
Management Considerations during OLT	Dissection Phase Highest risk for volume fluctuations due to hemorrhage Air entrainment risk while working on IVC Venous cannula in infra-hepatic venous circulation decompresses splanchnic venous pressure and reduces bleeding Adjust ECMO flows and volume based on hemodynamics and P_venous_ Maintain ECMO flows > 1 L/min Potential for air entrainment in ECMO circuit Anhepatic Phase Continued air entrainment risk Lactates are misleading for total body perfusion Neohepatic Phase Risk for vasoplegia, acute pulmonary artery pressure rise, and myocardial collapse ECMO circuit will reduce the need for aggressive pretreatment with inotropes and vasopressors immediately prior to reperfusion Titrate ECMO flow based on myocardial function and hemodynamics using TEE guidance in communication with perfusion
Emergent ECMO Circuit Failure	Have backup, primed ECMO circuit readily available Communicate with surgeon regarding periods of significant bleeding or potential IVC injury as this may necessitate temporary reduction in ECMO flow rates to reduce venous suction magnitude and ECMO entrainment risk

Adapted from the Mayo Clinic [8] in Florida ECMO Protocol with permission.

**Table 3 jcm-12-04965-t003:** Considerations for preemptive ECMO usage for liver transplantation [4,5,6,7,9,10,11].

Factors to Consider	Venous cannula in the IVC affects clamping of the IVC prior to the anhepatic stage for piggyback technique Anticoagulation for ECMO circuit can increase bleeding so consider using heparinase thromboelastograms to treat underlying coagulation profile Severe coagulopathy can be treated accordingly via targeted transfusion or factor replacement No anticoagulation increases risk of circuit thrombosis Have readily available, pre-primed second circuit in the event of emergent circuit failure Continuous renal replacement therapy (CRRT) can be utilized via direct connection to the ECMO circuit Large volume shifts will affect circuit flow and pump speed causing a risk of air entrainment and poor perfusion Optimize patient’s volume status prior to ECMO cannulation Patient will need postoperative ICU care and coordinative efforts with the critical care team should be established prior to surgery Timing for decannulation Multidisciplinary planning prior to surgery between transplant surgery, anesthesiology, critical care, cardiothoracic surgery, and perfusion
Intraoperative Management	Intracardiac thrombosis: establish treatment protocols prior to surgery. Whether or not anticoagulation and which kind will be given Small clots that do not cause hemodynamic instability → possibly no treatment, monitor with TEE Clots that are actively propagating and/or causing hemodynamic instability should be treated with heparin initially. If clot does not dissolve, consider Alteplase Large clots that cause complete cardiovascular collapse require immediate treatment with Alteplase Maintain ECMO flows > 1 L/min to minimize risk of thrombotic complication Have 2nd primed circuit readily available in the event of circuit failure Flows are to be titrated with TEE guidance Periods of hypovolemia necessitates decreased flow to minimize risk of entraining air into the circuit Decision to anticoagulate should be made on a case-by-case basis
Possible Postoperative Complications	Thrombotic complications (e.g., stroke, deep vein thrombosis, pulmonary embolism, etc.) Bleeding Inability to decannulate from ECMO

**Table 4 jcm-12-04965-t004:** Tools for the assessment of disability and frailty in liver transplant patients [16,17].

Exam	Grading
Activities of Daily Living (ADLs)	Grade ability to perform bathing, dressing, toileting, transferring, continence, eating, and transportation around area of residence. 1 point = independent 0 point = requiring assistance Severe: difficulty with 2 or more ADLs Moderate: difficulty with 1 ADL Mild: no difficulty with ADLs
Clinical Frailty Scale (CFS)	Scale of 1 (very fit) to terminally ill (9) 1: Very fit—robust, active, energetic, and motivated 2: Well—no active disease symptoms; active occasionally; independent 3: Managing well—medical problems well controlled but not regularly active; able to perform ADLs independently 4: Vulnerable—not dependent on others for ADLs but symptoms limit activities and easily tire 5: Mildly frail—evidently slower; need help with higher order ADLs (finances, transport, heavy housework, medications) 6: Moderately frail—need help with ALL ADLs 7: Severely frail—completely dependent for personal care 8: Very severely frail—completely dependent, approaching the end of life; would typically not recover from a minor illness 9: Terminally ill—life expectancy < 6 months
Karnofsky Performance Scale (KPS)	Grades function as percentage from 0% (deceased) to 100% (complete independence and thriving) OR Grades performance from A–C A (80–100%): minimal symptoms; normal activity B (50–70%): requires assistance from others and variable medical care C (0–40%): disabled; dependent on others for care
Fried Frailty Index (FFI)	Graded on performance of the following: - weight loss (<101 lbs in year) - exhaustion (>3 days/week) - hand-grip strength (controlled for gender and BMI) in the 20th percentile or less - 15 ft walk speed (controlled for gender and height) in the 20th percentile or less - activity on Minnesota Leisure-time Activity (controlled for gender) in the 20th percentile or less
6 min Walk and Distance	Meters walked in 6 min timed test
Short Physical Performance Battery (SPPB)	- Balance test with feet side-by-side, semi-tandem, and tandem - Hand-grip strength - Amount of time to rise from chair ×5 times
Gait Speed	Meters per second to walk set distance measured in meters/second
Cardiopulmonary Exercise Testing (CPET)	Patient placed on a stationary bike or treadmill with increasing workload while patient’s heart rate, blood pressure, and electrocardiographic monitoring is monitored. Patient wears a respirator while performing this exam which measures oxygen uptake, ventilation, and carbon dioxide output.

## Data Availability

Not applicable.

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
