# Peer review of "Preemptive Venoarterial Extracorporeal Membrane Oxygenation for Liver Transplantation—Judicious Candidate Selection"

_jcm, 2023, doi:10.3390/jcm12154965_

Round 1
Reviewer 1 Report
Manuscript entitled "Preemptive Venoarterial Extracorporeal Membrane Oxygenation for Liver Transplantation – Judicious Candidate Selection"
With a report and two cases with extensive discussion, the authors claimed that ECMO is a feasible option in high-risk patients. Multidisciplinary planning and careful consideration of associated complications with ECMO during LT is paramount.
I would suggest this work contains adequate information and could be accepted pending the following modifications:
1. Representative clinical images should be shown. For example, the CT images.
2. The authors are encouraged to add the timelines of these two patients to make the story easily understand.
Acceptable
Author Response
- Representative clinical images should be shown. For example, the CT images.
Echo images have been added as opposed to CT images since the paper highlights the use of ECMO in the setting of pulmonary hypertension a4 nd RV dysfunction. 4 intraoperative TEE images (2 baseline images and 2 images immediately after reperfusion) from patient 1 were added.
2. The authors are encouraged to add the timelines of these two patients to make the story easily understand.
The months and years of the two patients' hospital courses have been added to clarify the timeline.
Reviewer 2 Report
The authors have made an interesting attempt at “Preemptive Venoarterial Extracorporeal Membrane Oxygenation for Liver Transplantation – Judicious Candidate Selection.” The manuscript is interesting; however, the authors need to justify the scientific writing manuscript. Some of the general comments are provided below:
1. What was the study design used in this case report? Was it a single-center or multi-center study?
2. Did the study follow any specific guidelines or protocols for conducting liver transplant procedures and managing pulmonary hypertension?
3. How were the patient's ESLD and PPH diagnosed and assessed before the liver transplant?
4. What specific diagnostic criteria were used to determine the severity of pulmonary hypertension?
5. Were there any alternative treatment options considered for the patient's condition before proceeding with LT and VA ECMO?
6. To what extent do the findings and outcomes of this case report 1 apply to other patients with ESLD and PPH undergoing liver transplantation?
7. What were the specific parameters monitored during the intraoperative period? How were ECMO flows adjusted and managed?
8. Were there any unexpected events or complications during the transplantation procedure, and how were they addressed?
9. What were the reasons for administering intravenous heparin and calcium chloride during the procedure?
10. Were there any long-term follow-up assessments to evaluate the success and durability of the transplantation procedure?
11. What were the specific differences in the long-term postoperative outcomes between the two patients? Were there any complications or adverse events that contributed to these differences?
12. How representative are these two cases of the broader population of patients with high risk of right ventricular failure undergoing liver transplantation?
13. Have similar strategies been employed in other medical centers or studies, and if so, what were the reported outcomes?
14. Are there established protocols or algorithms for ECMO management during and after the LT procedure? If so, can you provide more details on their components?
15. How is the timing of ECMO decannulation determined, considering the risks of continued anticoagulation/hemostasis versus cardiopulmonary stability?
Author Response
- What was the study design used in this case report? Was it a single-center or multi-center study?
Single-center. We clarified in the manuscript.
2. Did the study follow any specific guidelines or protocols for conducting liver transplant procedures and managing pulmonary hypertension? No specific guideline/protocol for managing pulmonary hypertension. We do have a liver transplant protocol at our institution that we always follow. I clarified this in the manuscript.
3. How were the patient's ESLD and PPH diagnosed and assessed before the liver transplant? The patient's ESLD was diagnosed over several years in the outpatient setting by the medical transplant team via MELD score calculation and signs/symptoms. Patient 1 was diagnosed with ESLD when a new mass in 2021 was discovered to be hepatocellular carcinoma. Soon after, she began developing ascites and jaundice and was referred to our institution for a liver transplant workup. Patient 2 was diagnosed with cirrhosis around 2015-2016 when he was found to have low platelets during routine labs. Both patients; PPH was diagnosed by serial TTEs during their hospital course and by right heart cath as part of the pretransplant workup.
4. What specific diagnostic criteria were used to determine the severity of pulmonary hypertension? We used the diagnostic criteria outlined in the beginning of the introduction to determine the severity of pulmonary hypertension in this case series. I clarified this in the manuscript.
5. Were there any alternative treatment options considered for the patient's condition before proceeding with LT and VA ECMO? The only alternative was to not proceed with transplantation as the high-risk committee deemed both patients too high risk with pressors/inotropes and invasive monitoring alone.
6. To what extent do the findings and outcomes of this case report 1 apply to other patients with ESLD and PPH undergoing liver transplantation? The implications of case report 1 that we wanted to highlight is that even though a high-risk PPH patient can survive the immediate intraoperative period with ECMO support, the bigger question is whether we should even do that if the patient is not likely to survive the postoperative period. This concept applies to other ESLD, PPH patients undergoing liver transplantation. We want readers to consider the patient’s long-term survivability and rehabilitation potential in addition to just the immediate intraoperative survivability should they encounter this type of patient.
7. What were the specific parameters monitored during the intraoperative period? How were ECMO flows adjusted and managed?
Both cases were performed with invasive monitoring including an arterial line, central venous catheter, pulmonary artery catheter, rapid infusion catheter and TEE. The ECMO flows were adjusted and managed under TEE guidance and based on the patient's hemodynamics intraop. I also further clarified this in the manuscript.
8. Were there any unexpected events or complications during the transplantation procedure, and how were they addressed? Fortunately, the problems that arose during surgery (e.g. anticoagulation, cannula positioning, fluid shifts, etc.) were anticipated and discussed during the multiple multidisciplinary discussions prior to surgery. Contingency plans and even a second circuit were readily available in case of these complications. We talk about this in the discussion section and table 2 which outlines the solutions to complications we anticipated.
9. What were the reasons for administering intravenous heparin and calcium chloride during the procedure? IV heparin was administered to prevent thrombotic complications while on ECMO and calcium was administered for inotropic support. This was clarified in the manuscript.
10. Were there any long-term follow-up assessments to evaluate the success and durability of the transplantation procedure? All transplant patients at our institution are followed closely in the outpatient setting by medical transplant team. They are seen in clinic 6 months post transplant, one year post transplant, and annually afterwards. Labs and ultrasounds are obtained on these follow up visits to ensure adequate graft performance. Patient 1 passed during her postop ICU course.
11. What were the specific differences in the long-term postoperative outcomes between the two patients? Were there any complications or adverse events that contributed to these differences?
Patient 1 passed away in the ICU during her postop hospital course and was never able to be downgraded to the floor. Patient 2 was able to be successfully discharged from the ICU to the floor and eventually to outpatient rehab. The major postop complications due to the ECMO cannulas in patient 2 was narrowing at the venous piggyback anastomosis. This was corrected by balloon angioplasty. Patient 2 and the liver graft did well afterwards. Patient 1's hospital course was complicated by worsening hypoxia, continued hemodynamic instability necessitating increasing inotropic and vasopressor support, worsening pulmonary artery pressures requiring increasing veletri and inhaled nitric oxide, and coagulopathy. She progressively declined throughout her ICU stay and eventually passed away from supra-systemic pulmonary pressures. This is in the manuscript.
12. How representative are these two cases of the broader population of patients with high risk of right ventricular failure undergoing liver transplantation? These two cases are a good representation of the liver transplant patients with pulmonary hypertension. PPH is not an uncommon sequelae of end stage liver disease and this comorbidity makes a patient significantly more high risk intraoperatively especially during the reperfusion stage when the RV experiences an onslaught of previously ischemic volume upon cross clamp removal. This is the most dangerous portion of the surgery and RV collapse is the top concern. Most of the time, these patients would NOT be a surgical candidate due to this risk. We present that ECMO alleviates this problem and allows us to bridge the patient with pulmonary hypertension past this point despite a weak RV allowing patients that are previously not transplant candidates to undergo surgery.
13. Have similar strategies been employed in other medical centers or studies, and if so, what were the reported outcomes? The preemptive use of ECMO for high-risk pulmonary hypertension patients for liver transplantation has been supported in the literature. A Duke case report presents a successful liver transplantation on a high risk pulmonary hypertension patient with preemptive initiation of VA-ECMO after previous aborted attempts. A high-volume transplant center with an extensive database of liver transplant patients on ECMO present ECMO as a viable rescue strategy that should be considered in patients with severe cardiopulmonary compromise. I included this in the manuscript.
14. Are there established protocols or algorithms for ECMO management during and after the LT procedure? If so, can you provide more details on their components? Yes, we have an institution protocol for liver transplant on ECMO which we go over in table 5.2 and in the discussion protocol. We do not have a post-liver transplant ECMO protocol once the patient is in the ICU as this would fall under the jurisdiction of the cardiothoracic ICU who regularly manage ECMO and other mechanical circulatory devices.
- How is the timing of ECMO decannulation determined, considering the risks of continued anticoagulation/hemostasis versus cardiopulmonary stability? This decision is a multidisciplinary discussion made between transplant ICU, cardiac surgery, and the medical transplant team. The patient is progressively weaned off ECMO in the ICU and a trial off period is performed to prove that the patient is able to maintain life support through conventional means. Since these two patients were NOT dependent on ECMO prior to transplantation but were instead cannulated to support their cardiac function during the extreme stress of liver transplantation, ECMO decannulation was able to be conducted within a few days after surgery once the stressor had been removed. Repeat echocardiography is also performed to ensure adequate cardiac recovery and function prior to decannulation.
Round 2
Reviewer 2 Report
The authors have modified the manuscript and now it is acceptable for publication.